# Perceptions and Feasibility of Actions Related to Sodium Reduction among Restaurant Owners and Cooks in Seongnam, South Korea: Comparison According to Stages of Behavioral Change

**DOI:** 10.3390/nu13124375

**Published:** 2021-12-06

**Authors:** So-Hyun Ahn, Jong-Sook Kwon, Kyungmin Kim, Hye-Kyeong Kim

**Affiliations:** 1Department of Food and Nutrition, Shingu College, Seongnam 13174, Korea; sohyunahn1123@nate.com (S.-H.A.); jskwon@shingu.ac.kr (J.-S.K.); 2Department of Food and Nutrition, Baewha Women’s University, Seoul 03039, Korea; kyungmkim@baewha.ac.kr; 3Department of Food Science and Nutrition, The Catholic University of Korea, Bucheon 14662, Korea

**Keywords:** stages of change, restaurants, sodium reduction, feasibility, transtheoretical model

## Abstract

With the increase in meals eaten outside the home, sodium reduction in restaurant foods is essential for reducing sodium intake. This study aimed to assess the stages of behavioral change for reducing sodium and the differences in perceptions among restaurant staff by stage. Restaurant owners and cooks (*n* = 313) in Seongnam, South Korea were surveyed on their stage of behavioral change, practices, and perceptive factors related to sodium reduction in restaurant meals using a questionnaire. The proportion of behavioral change by stage was 20.4% in the maintenance and action (MA) stage, 32.3% in the preparation (P) stage, and 47.3% in the pre-preparation (PP) stage, which included contemplation and pre-contemplation stages. The items that represent differences among the groups were recognition of social environment for sodium reduction, practice of weighing condiments and measuring salinity, and feasibility of actions related to low-sodium cooking. Logistic regression analysis was used to estimate odds ratios for practice and perceptive factors by using stage of behavioral change as the independent variable. Factors associated with being in the MA stage were weighing condiments, measuring salinity, and high feasibility of actions related to low-sodium cooking. Recognition of sodium labeling and anticipation of better taste by reducing sodium increased the odds of being in the P stage rather than the PP stage. These results suggest that customized stepwise education and support are needed for the efficacy of restaurant-based sodium reduction programs.

## 1. Introduction

High sodium intake is strongly associated with hypertension, a major metabolic risk factor leading to cardiovascular diseases and death throughout the world [1,2]. High dietary sodium intake is also associated with an increased risk of stomach cancer [3] and has been shown to adversely affect the heart, kidneys, brain, and bone [4]. In this regard, the World Health Organization (WHO) has set a goal of a 30% relative reduction in population intake of sodium by 2025 and established a target for an adult sodium intake of less than 2000 mg/day [5]. It has been reported that global mean sodium intake in 2010 was 3.95 g/day [6], nearly twice the WHO recommended limit. Many countries have taken action to achieve the sodium reduction goal [7]. In 2012, South Korea began implementing the National Movement to Reduce Sodium Intake [8]. Consequently, the sodium intake of Koreans has shown a decreasing trend. The daily sodium intake was 3286 mg in 2019, a more than 25% decrease compared with 4789 mg in 2010 [9]. However, sodium intake remains much higher than the recommended daily upper limit.

The sources of dietary sodium vary by region. In Europe and Northern countries, most dietary sodium comes from sodium added in manufactured foods (approximately 75% of the total intake), while salt and soy sauce added during food preparation and at the table are the primary sources in East Asian countries [10,11,12]. With the increased consumption of meals cooked outside the home, contribution of restaurant food to dietary sodium intake has become significant. It was reported that restaurant meals had the highest sodium density (per 1000 kcal) among food sources in the United States [13]. In South Korea, sodium intake from restaurant meals was 1.5-fold higher than that from home-cooked meals [14]. Furthermore, the rate of eating out at least once per day in South Korea increased to 33.3% in 2019 [9]. Therefore, sodium reduction in restaurants should be an essential strategy to achieve the sodium reduction goal.

Many countries have implemented policies to reduce sodium in restaurant food, focusing primarily on chain restaurants through menu labeling and target setting for reducing sodium in dishes [15]. Similarly, in South Korea, the government promoted a menu consulting project and restaurants have been encouraged to join a voluntary ‘healthier restaurants’ program, which designates and publicizes restaurants that offer a considerable number of low-sodium meals or sodium-reduced menus [16]. Some local governments have developed customized programs to help reduce sodium in restaurant foods, such as low-sodium cooking consultation and training in the use of the salimeter and a mobile application for salinity management. The programs had a positive effect on reducing sodium content of restaurant menu items [17]. However, it has not been investigated what factors drive changes in reducing sodium in restaurant food.

Restaurant owners and cooks play a crucial role in making healthy foods available, suggesting the importance of their willingness to change in order to promote the reduction of sodium in their restaurant. Programs planned and implemented on a theoretical basis can be both effective and practical [18]. The transtheoretical model is useful for assessing whether restaurant owners and cooks are oriented toward change and the establishing of effective strategies. According to the model, people do not make changes all at once but move through five behavioral stages when adopting a health-related behavior: pre-contemplation, contemplation, preparation, action, and maintenance [19]. In addition, social cognitive theory has been widely applied in understanding health-related behaviors and behavior modification [20]. The theory states that behavior is a function of environmental factors and personal factors including outcome expectancy of the behavior, skills and knowledge necessary to perform the behavior, and self-efficacy. Therefore, it is necessary to understand the relationship between the stages of behavioral change and perceptions of factors related to practices for developing effective intervention programs. Despite the critical roles of restaurant owners and cooks, there has been little research on their perception of sodium reduction and their readiness to reduce sodium in restaurant foods.

This study aims to assess the status of behavioral change in reducing sodium among restaurant owners and cooks in South Korea and characterize perceptions and the feasibility of actions by stage to provide suggestions for effective sodium reduction in restaurants.

## 2. Subjects and Methods

### 2.1. Study Design and Participants

A cross-sectional survey was performed to investigate the stages of behavioral change, current practice, and perceptions related to reducing sodium in restaurant owners and cooks. Participants were recruited via mandatory annual hygiene training for restaurants, which was hosted by the Food Safety Department of Seongnam City, South Korea. The participants attended the training program on behalf of the restaurant for which they worked. A total of 315 people signed written consent forms for inclusion after hearing the purpose and content of the survey. This study was approved by the Institutional Review Board of Shingu College (2017–101). The survey was conducted from May to July 2017.

### 2.2. Questionnaire

A self-administered questionnaire was developed, with modifications, from those used in previous studies [21,22,23]. The questionnaire consisted of three sections: general information on participants and the restaurants in which they worked, questions for classifying the stages of behavioral change, and practices and perceptive factors related to reducing sodium in restaurant foods. The survey questionnaire is presented in Appendix A.

#### 2.2.1. General Information on Participants and Restaurants

Participant information included age, gender, and position in the restaurant. They were also asked questions about the restaurant where they worked, such as size, location, operating system, and the type of food served.

#### 2.2.2. Stage of Behavioral Change

The stage of behavioral change for reducing sodium in the restaurant food was assessed using an algorithm with separate questions (Figure 1). Participants were classified into five stages: maintenance, action, preparation, contemplation, and pre-contemplation.

#### 2.2.3. Practices and Perceptive Factors Related to Reducing Sodium in Restaurant Food

The current status of practicing sodium reduction in the restaurant was examined using four criteria: the presence of salt or salty condiments in the table, the frequency of weighing salt and salty condiments while cooking, the frequency of measuring food salinity, and providing salinity information for the restaurant food. Participants were asked to respond ‘almost always,’ ‘often,’ or ‘scarcely’ to the frequency questions. Perceptive factors related to reducing sodium in restaurant food were investigated within three categories: recognition of the social environment, positive outcome expectancies and barriers in practicing sodium reduction, and the feasibility of actions related to sodium reduction in restaurants. The list of positive outcome expectancies and barriers in reducing sodium in restaurant food was suggested, and participants were asked to select the main factors. The feasibility of actions related to sodium reduction in restaurants was evaluated with 15 questions, classified into five sections: purchase, cooking, menu adjustment, serving, and offering information on sodium reduction. The feasibility of the actions was rated using a 5-point Likert scale ranging from 1 (very easy) to 5 (very difficult). Cronbach’s alpha coefficient for the feasibility of action items was 0.83, indicating high internal consistency in the responses. In addition, the participants were asked to indicate their willingness to participate in sodium reduction under the government’s support and select support needs for practicing sodium reduction in restaurant food. The list of support was suggested based on the policy of the Healthy Restaurant for Sodium Reduction program initiated by the Korean Ministry of Food and Drug Safety.

### 2.3. Statistical Analysis

All statistical analyses were performed using the SAS (version 9.3, SAS Institute, Inc., Cary, NC, USA) software package. Age and Likert scores were reported as means ± standard deviations (SD). The possible confounders in this study are age, gender, and position of participants in restaurant, types of food served, and size and operating system of restaurant. The means of numerical data were compared by analysis of covariance (ANCOVA), and categorized data were analyzed using Chi-square tests, according to the stage of behavioral change. Multivariate logistic regression was used to investigate the relationship between the stage of change and practices and perceptive factors regarding sodium reduction in restaurant food. Results are presented as odds ratio (OR) and 95% confidence interval (95% CI). A *p*-value of < 0.05 was considered statistically significant.

## 3. Results

### 3.1. Participants and Restaurants Information

A total of 313 participants were included in the study after excluding people who did not answer the question for assessing the stage of behavioral change (inclusion rate 99%). Table 1 shows the characteristics of participants and restaurants. The mean age of the participants was 49.1 years old (SD = 10.4), and the proportion of men was 59.4%. The participants were restaurant owners (43.2%), cooks (6.1%), or worked as the owner and cook (50.7%), which indicates that they were aware of most cooking practices. The most common food served was home-style Korean meals, including cooked rice and side dishes (37.9%). More than 90% of restaurants were medium or small-sized restaurants of less than 300 m^2^. The size and regional distribution of the restaurants surveyed were not very different from Seongnam City’s restaurant data [24]. Approximately three-fourths of the restaurants were independently operated. When participants were classified according to the stage of behavioral change, the proportion of participants in the maintenance stage, action stage, and preparation stage were 16.6%, 3.8%, and 32.3%, respectively. However, almost half of the participants were in the contemplation stage (31.3%) and pre-contemplation stage (16.0%). These two stages were combined for analysis because understanding the characteristics of participants in these stages is necessary for progress in making changes. In addition, the maintenance stage and action stage were combined because the number of participants in the action stage was relatively small. Thus, the five stages of behavioral change were reduced to three groups: maintenance and action (MA), preparation (P), and pre-preparation (PP). The age of the participants and the operating system were significantly associated with the stage of behavioral change. Participants in the MA group were older than the other groups, and the proportion of restaurants operated independently was lower in the PP group than in the other groups. Thus, analyses of the association between perceptive factors and stage of behavioral change were adjusted by the age of participants and the operating system of restaurants.

### 3.2. Recognition of Social Environment and Current Practices Regarding Sodium Reduction in Restaurant Food

Table 2 shows the results for recognizing the social environment and current practices regarding sodium reduction in restaurant food. The percentages of restaurant owners and cooks who recognized social efforts to reduce sodium intake through media campaigns differed significantly between the three groups (*p* = 0.0071). The percentages were 96.9%, 88.1%, and 81.1% in the MA, P, and PP groups, respectively. Similarly, the proportion of owners and cooks recognizing sodium content labeling on foods in restaurant or highway rest area was descending in the order of MA (76.6%), P (64.0%), and PP (49.0%). Recognizing sodium content labeling on food service industries increased the odds of being in the P stage rather than the PP stage by 1.85-fold (95% CI; 1.10–3.12).

Approximately 80% of restaurants had removed table salt. The practice of weighing salt and salty condiments while cooking differed between the groups (*p* = 0.0165). The percentage of participants who reported almost always weighing salt and salty condiments was 65.6%, 49.5%, and 44.2% in the MA, P, and PP groups, respectively. Weighing salt and salty condiments while cooking almost always increased the odds of being in the MA stage rather than the P stage (OR; 1.95, 95% CI; 1.02–3.72). The practice of measuring food salinity was also different according to the stage of behavioral changes (*p* = 0.0068). The rate was the highest in the MA group, followed by the P group and PP group. Participants who measured the salinity of food always or often had a higher odds ratio of being in the MA stage versus the P stage (OR; 2.16, 95% CI; 1.07–4.37).

### 3.3. Positive Outcome Expectancies and Barriers in Reducing Sodium

Table 3 presents the results on positive outcome expectancies and barriers in reducing sodium in restaurant food. Restaurant owners and cooks perceived ‘better health of customers and employees’ as the primary positive outcome. Approximately half of owners and cooks believed that it would help ‘improve the restaurant’s image’. Only the item of ‘better taste by revealing the natural flavor in food’ showed a significant difference in agreement between the groups (*p* = 0.0382). The proportion of participants in agreement with ‘better taste by revealing the natural flavor in food’ was the highest for those in the P stage, followed by the MA stages and PP stage. Owners and cooks expecting the item as a benefit of reducing sodium use had higher odds of being in the P stage versus the PP stage (OR; 1.96, 95% CI; 1.17–3.30). Among the barriers in practicing sodium use reduction, ‘hard to maintain taste’ was chosen as the primary factor, with more than 60% of owners and cooks agreeing. ‘High cost’ and ‘short shelf-life and spoilage of foods’ were the barriers that showed differences in agreement between the groups (*p* < 0.05). The proportions of owners and cooks who acknowledged these barriers were lowest in the MA group. In addition, having ‘limited knowledge and skills to practice’ and ‘short shelf-life and spoilage of foods’ barriers reduced the odds of a participant being in the MA stage versus the P stage by less than half (0.47 and 0.28, respectively). This result suggests that overcoming these barriers is important when taking actions to reduce sodium use in restaurants.

### 3.4. Feasibility of Actions Regarding Sodium Reduction among Restaurant Owners and Cooks

Table 4 presents the differences in the feasibility of actions regarding sodium reduction according to the stage of behavioral change. The total average score and the average score of the cooking section were significantly different between the groups. The scores were lower in the MA group compared with other groups, indicating that restaurant owners and cooks in the MA stage have a higher probability of practicing actions related to sodium reduction. When multivariate logistic regression was performed, owners and cooks who answered that it would be easy or very easy to practice actions such as ‘use salimeter and maintain standard salinity’, ‘cook with low-sodium sauce or broth’, ‘use herbs and spices to flavor dishes’, and ‘apply cooking skills to make low-sodium dishes’ were more than twofold likely to be in the MA stage. ‘Putting up promotional materials for reducing sodium intake’ was also related to being in the MA stage rather than the P stage, which indicates the confidence of practicing low-sodium meal preparation in the MA group (OR; 2.09, 95% CI; 1.11–3.95). ‘Serving sauce separately’ was the only item related to a participant being in the P stage rather than the PP stage (OR; 1.91, 95% CI; 1.13–3.24), indicating that it is a feasible item for owners and cooks to attempt with an intention to reduce sodium.

### 3.5. Support Needs to Reduce Sodium in Restaurant Food

Table 5 shows the support needed to reduce sodium use in restaurants. More than 80% of owners and cooks answered that they would participate in sodium reduction under the support of the government. The proportion was the highest in the P group, followed by the MA and PP groups. Owners and cooks willing to participate in sodium reduction with government support had higher odds of being in the P stage versus the PP stage (OR; 2.41, 95% CI; 1.15–5.04), indicating that support could be effective for owners and cooks to consider reducing sodium in restaurant food. Among the items of support needs for practicing sodium reduction, ‘providing plaques of healthy restaurant to participating restaurant’ was selected the most often, followed by ‘training cooking skills to reduce salt use’ and ‘providing salimeter and educating how to use.’ There were no differences in the proportion of support needs between the groups. When multivariate logistic regression analysis was performed, the support needs of ‘analyzing sodium content and support to display it’ was the only item to show significance; it reduced the odds of being in the MA stage versus the P stage by less than half (OR; 0.47, 95% CI; 0.24–0.93).

## 4. Discussion

The present study provides the status of behavioral changes for reducing sodium use among restaurant owners and cooks in South Korea. South Korea implemented the National Movement to Reduce Sodium Intake in 2012, with multisectoral intervention targeting the food industry, consumers, meal service, restaurants, and media. Sodium intake in South Korea decreased more than 20% over four years, accompanied by reductions in population blood pressure and hypertension prevalence [16]. The food industry successfully reformulated many processed foods to contain less sodium, and continuous nationwide campaigns created public awareness [16]. Our previous study showed that the percentages of Korean consumers in each stage of behavioral change for sodium reduction were 29.5% in the maintenance stage, 19.5% in the action stage, and 3.7% in the preparation stage [25]. Compared to consumers, a lower proportion of restaurant owners and cooks were classified in the maintenance and action stages (20.4%), but a higher proportion were classified in the preparation stage, indicating that it is not easy to find low-sodium foods in restaurants, despite half of consumers already practicing sodium reduction. Similarly, a previous survey on restaurant owners and chefs working in Philadelphia Chinese take-out restaurants reported that only 18% offered low-sodium options on their menus, while more than 40% had been asked by customers about low-sodium options [22]. In this study, about one-third of restaurant owners and cooks were classified in the preparation stage, in which sodium reduction is not practiced, but there is the intention to make changes in the near future. Moreover, almost half of restaurant owners and cooks were classified in the pre-preparation stage in which motivation is required to make changes. Thus, it is necessary to develop stage-specific strategies for restaurant owners and cooks to address their differing needs.

We examined the characteristics of perceptive factors and the feasibility of actions by stages based on social cognitive theory to assess the factors associated with motivation for making changes and taking actions to reduce sodium in restaurant food. The restaurant owners and cooks were classified into three groups according to behavioral stages: MA, P, and PP. Multivariate logistic regression analysis showed the significant factors differentiating the three groups. The comparison of respondents in the P group with those in the PP group was considered as determining the odds of ‘being motivated for making changes to reduce sodium,’ while the comparison of respondents in the MA group with those in the P group revealed the odds of ‘taking action to reduce sodium intake.’

The perception of the social environment related to reducing sodium can be an important compelling factor for sodium reduction in restaurants. More than 80% of restaurant owners and cooks recognized the social efforts to reduce sodium intake through nationwide campaigns and publicity. Nonetheless, a higher proportion of owners and cooks in the MA stage suggests the need for increasing awareness. Recognizing sodium content labeling on foods in restaurants or highway rest areas was a meaningful factor for being motivated to make changes to reduce sodium. Thus, it could be an effective way of motivating restaurant owners and cooks, by informing them that food service industries are already trying to reduce sodium. In addition, the significant differences in practice rates of weighing salt and salty condiments and measuring salinity while cooking between the groups indicate the importance of these actions in reducing sodium in restaurants. Furthermore, these factors were associated with a high probability of taking actions beyond the preparation stage. While the amount of sodium in most western foods primarily comes from standardized ingredients, the amount of sodium in Asian foods is largely determined by cooks through seasoning addition while cooking [26,27]; this finding indicates why the comprehension of perception and the behavioral stage of reducing sodium among restaurant staff is an important factor for reducing sodium in restaurant foods in Asian countries. Therefore, weighing salty condiments and measuring salinity are significant guidelines for practicing sodium reduction in restaurants.

Social cognitive theory suggests that essential factors related to changes in behavior are outcome expectancies, self-efficacy, and environmental factors [20]. Outcome expectancies are important for motivation to elicit behavior change. High expectation of better health when practicing sodium reduction is consistent with a previous study of restaurant staff [22]. It is noticeable that the expectation of better taste by revealing the natural flavor in food was associated with the likelihood of being motivated to make changes to reduce sodium, considering that concerns about maintaining taste and customer complaints were the main barriers in practicing sodium reduction. In addition, the restaurant owners and cooks who perceived ‘limited knowledge and skills to practice sodium reduction’ as a main barrier were more likely to be in the P stage rather than the MA stage. These results indicate that the expectation of better taste by reducing sodium could be motivational, but knowledge and skills related to cooking are needed for implementation of sodium reduction.

Perceived self-efficacy is strongly related to the capability to change behavior, which requires knowledge and skills to manage prospective situations [28]. In this study, restaurant owners and cooks were asked about the feasibility of actions instead of individual self-efficacy in order to assess the practicability of sodium reduction in their restaurant. The differences in total average score and the average score of the cooking section between the groups were as expected from the result on barriers. Practicing sodium reduction was more likely among owners and cooks who believed that it would not be too difficult to execute specific actions for reducing sodium while cooking, which also suggests the need for cooking skills training for motivated owners and cooks to attempt to execute sodium reduction. Among the items in feasibility assessment, ‘purchase foods after comparing sodium content in nutrition labels’ was the hardest action to perform, which corresponds with a previous report that indicated only 22% of restaurant staff used nutrition labels when selecting foods for restaurants [22]. Thus, along with the availability of low-sodium food, training in food procurement is needed to reduce sodium in restaurant meals.

It is encouraging that most owners and cooks showed a willingness to participate in sodium reduction under the support of the government, and the willingness was associated with being motivated to make changes. Thus, it will be a good strategy to publicize the support of the government for motivating restaurant owners and cooks to participate in sodium reduction. While a previous survey on Chinese restaurant owners and chefs in the United States reported a need for training in food purchase, preparation, and marketing for low-sodium dishes [22], this study showed a high demand for restaurant promotion, such as providing award plaques to healthy restaurants and publicizing participating restaurants, in addition to technical training. This finding was supported by the result that almost half of the restaurant owners and cooks expected that sodium reduction would improve the restaurant’s image. Concerns about customer satisfaction and maintaining revenue were reported to be major barriers to providing low-sodium menus at worksite cafeterias [29], which seems to be more pronounced in restaurants serving unspecified customers. Moreover, widely used strategies to reduce sodium, such as reformulation and development of new dishes, are challenging and cost time and money. Restaurant owners are not willing to offer healthier food options unless it can increase profits [30], which explains that there has been little to no change in mean sodium levels in restaurant foods despite salt reduction interventions [31]. Therefore, other aspects need to be considered when developing strategies for restaurants. An intervention study in Korea showed that customers were satisfied with sodium-reduced menus when they were aware that the restaurants were participating in a sodium reduction program [30]. In addition, restaurant owners and cooks who consumed less dietary sodium were more willing to participate in low-sodium restaurant initiatives [32]. These studies imply that raising awareness of a healthy diet by consumers and restaurant workers remains important. However, few strategies exist that focus on education in restaurant policies for reducing sodium.

To our knowledge, this is the first study to assess the stages of behavioral changes among restaurant owners and cooks with regard to sodium reduction and to describe the differences in perceptions and feasibility of actions by stage. However, this study has some limitations. First, the survey was conducted in a limited region in South Korea and may not reflect restaurant owners and cooks in South Korea as a whole. Second, the response of restaurant owners and cooks may be biased toward willingness and action for reducing sodium in order to meet social expectation, which suggests the need for increased intervention. Additionally, self-reported practices in a restaurant may not be as accurate as observation. Nonetheless, the responses revealed the differences in practice status and perceptions according to the stages. Third, the questionnaire did not include knowledge items related to sodium intake, as we focused on perception and procedural actions associated with reducing sodium in a short questionnaire. Future studies are needed to present detailed educational content for restaurant workers.

## 5. Conclusions

Although consideration of restaurant meals is an integral part of reducing sodium intake, only about 20% of restaurant owners and cooks in South Korea were in the action and maintenance stages for reducing sodium use in their restaurants. It is necessary to plan strategies to assist in promoting motivation for making changes and putting changes into practice. The differences in perceptive factors and feasibility of actions according to stages indicate that recognition of sodium reduction trends in the food service industry and anticipation of better taste can be motivating factors for making changes. Still, knowledge and skills related to cooking are required for taking action to reduce sodium. Therefore, stepwise customized education and training will be needed to improve the efficacy of restaurant-based sodium reduction programs, along with support for participating restaurants and the raising of public awareness of a healthy diet.

## Figures and Tables

**Figure 1 nutrients-13-04375-f001:**
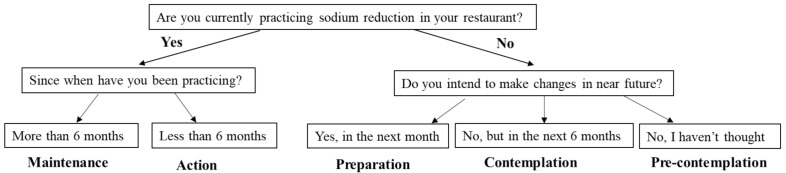
Classification of the stage of behavioral change for sodium reduction in restaurant food.

**Table 1 nutrients-13-04375-t001:** General characteristics of participants and restaurants.

	MA (*n* = 64)	P (*n* = 101)	PP (*n* = 148)	Total (*n* = 313)	*p*-Value ^1^
Stage of change (%)	20.4	32.3	47.3	100.0	
Age	54.5 ± 8.2 ^2a^	48.5 ± 10.6 ^b^	48.0 ± 10.9 ^b^	49.1 ± 10.4	0.0113
Age					
Under 40	6 (9.4) ^3^	20 (19.8)	34 (23.0)	60 (19.2)	0.0158
40–49	14 (21.9)	19 (28.7)	50 (33.8)	93 (29.7)	
50–59	30 (46.9)	42 (41.6)	42 (23.4)	114 (36.4)	
Over 60	14 (21.9)	10 (9.9)	22 (14.9)	46 (14.7)	
Gender					
Male	35 (57.8)	69 (68.3)	80 (54.0)	186 (59.4)	0.0761
Female	27 (42.2)	32 (31.7)	68 (46.0)	127 (40.6)	
Position in restaurant					
Owner	27 (42.9)	36 (35.6)	71 (48.6)	134 (43.2)	0.3501
Cook	4 (6.4)	8 (7.9)	7 (4.8)	19 (6.1)	
Owner and cook	32 (50.8)	57 (56.4)	68 (46.6)	157 (50.7)	
Location of restaurant					
Bundang-gu	80 (54.1)	41 (41.0)	25 (39.1)	146 (46.8)	0.1879
Sujeong-gu	19 (12.8)	17 (17.0)	10 (15.6)	46 (14.7)	
Jungwon-gu	49 (33.1)	42 (42.0)	29 (45.3)	120 (38.5)	
Types of food served					
Korean meal (cooked rice and side dishes)	32 (51.6)	37 (37.0)	48 (32.6)	117 (37.9)	0.2398
Grilled meat	5 (8.1)	9 (9.0)	15 (10.2)	29 (9.4)	
Japanese food	8 (12.9)	8 (8.0)	13 (8.8)	29 (9.4)	
Chinese food	2 (3.2)	6 (6.0)	11 (7.5)	19 (6.2)	
Western food	4 (6.5)	3 (3.0)	7 (4.8)	14 (4.5)	
Snack food	1 (1.6)	10 (10.0)	9 (6.1)	20 (6.5)	
Burger, pizza, fried chicken	4 (6.5)	6 (6.0)	18 (12.2)	28 (9.1)	
Bakery	1 (1.6)	4 (4.0)	2 (1.4)	7 (2.3)	
Others	5 (8.1)	17 (17.0)	24 (16.3)	46 (14.9)	
Size of restaurant					
Under 100 m^2^	40 (63.5)	61 (61.6)	86 (60.6)	187 (61.5)	0.2564
100~300 m^2^	19 (30.2)	27 (27.3)	50 (35.2)	96 (31.6)	
Over 300 m^2^	4 (6.3)	11 (11.1)	6 (4.2)	21 (6.9)	
Operating system of restaurant					
Independently owned	48 (75.0)	81 (80.0)	96 (64.9)	225 (71.9)	0.0251
Chain restaurant	16 (25.0)	20 (20.0)	52 (35.1)	88 (28.1)	

^1^ *p*-value from chi-square test or ANOVA according to the stage of change. ^2^ Mean ± SD, Mean values with different superscripts (a, b) are significantly different among the groups at α = 0.05 as determined by Duncan’s multiple range test after one-way ANOVA. ^3^
*N* (%). MA; Maintenance and Action stages, P; Preparation stage, PP; Pre-Preparation stage (contemplation and precontemplation stages).

**Table 2 nutrients-13-04375-t002:** Recognition of social environment and practices regarding sodium use in restaurant according to the stages of behavioral change.

	MA (*n* = 64)	P (*n* = 101)	PP (*n* = 148)	Total (*n* = 313)	*p*-Value ^1^	P to Reference PP		MA to Reference P	
						OR ^2^	95% CI	OR	95% CI
Recognition of social efforts for reducing sodium intake									
Yes	62 (96.9) ^3^	89 (88.1)	120 (81.1)	271 (86.6)	0.0071	1.73	0.83–3.59	4.18	0.90–19.3
Recognition of sodium labeling in restaurant or highway rest area									
Yes	49 (76.6)	64 (64.0)	72 (49.0)	185 (59.5)	0.0005	1.85 *	1.10–3.12	1.84	0.91–3.73
Removal of table salt									
Yes	62 (81.3)	82 (81.2)	116 (78.4)	250 (79.9)	0.8229	1.19	0.63–2.25	1.00	0.45–2.24
Weighing salt and salty condiments while cooking									
Almost always	42 (65.6)	50 (49.5)	65 (44.2)	157 (50.3)	0.0165	1.24	0.74–2.06	1.95 *	1.02–3.72
Measuring salinity of food									
Almost always or often	23 (36.5)	21 (21.0)	25 (16.9)	69 (22.2)	0.0068	1.31	0.69–2.49	2.16 *	1.07–4.37
Providing salinity information									
Yes	12 (19.1)	9 (8.9)	13 (8.9)	34 (11.0)	0.0517	1.00	0.41–2.44	2.45	0.97–6.22

^1^ *p*-value from chi-square test according to the stage of change. ^2^ OR; Odds Ratio, CI; Confidence Interval. * Independently significant in multivariate logistic regression models adjusted for age and operating system of restaurant (*p* < 0.05). ^3^ *N* (%). MA; Maintenance and Action stages, P; Preparation stage, PP; Pre-Preparation stage (contemplation and precontemplation stages).

**Table 3 nutrients-13-04375-t003:** Positive outcome expectancies and barriers in practicing sodium reduction in restaurants according to the stages of behavioral change.

	MA (*n* = 64)	P (*n* = 101)	PP (*n* = 148)	Total (*n* = 313)	*p*-Value ^1^	P to Reference PP		MA to Reference P	
						OR ^2^	95%CI	OR	95% CI
Positive outcome expectancies									
Improve the restaurant’s image	30 (46.9) ^3^	47 (46.5)	76 (51.4)	153 (48.9)	0.7093	0.83	0.50–1.37	1.01	0.54–1.90
Better health of customers and employees	50 (78.1)	79 (78.2)	114 (77.0)	243 (77.6)	0.9704	1.07	0.58–1.97	0.99	0.47–2.12
Better taste by revealing the natural flavor in food	25 (39.1)	49 (48.5)	48 (32.4)	122 (39.0)	0.0382	1.96 *	1.17–3.30	0.68	0.36–1.29
Satisfy customers’ varied preference for saltiness	25 (39.1)	46 (45.5)	62 (41.9)	133 (42.5)	0.6994	1.16	0.70–1.93	0.76	0.41–1.45
Help to develop menu by having more interest in cooking method	21 (32.8)	33 (32.7)	55 (37.2)	109 (34.8)	0.7130	0.82	0.48–1.40	1.01	0.52–1.96
Barriers to practice									
Hard to maintain taste	38 (59.4)	60 (59.4)	94 (63.5)	192 (61.3)	0.7564	0.84	0.50–1.41	0.99	0.53–1.89
Time-consuming and inconvenient process of cooking	27 (42.2)	52 (51.5)	77 (52.0)	156 (49.8)	0.3884	0.98	0.59–1.62	0.69	0.36–1.29
Limited knowledge and skills to practice	11 (17.2)	31 (30.7)	42 (28.4)	84 (26.8)	0.1368	1.12	0.64–1.94	0.47 *	0.22–1.02
Limitation to choose food items for low- sodium dishes	28 (43.8)	45 (44.6)	64 (43.2)	137 (43.8)	0.9792	1.06	0.63–1.76	0.97	0.52–1.82
High cost	8 (12.5)	22 (21.8)	44 (29.7)	74 (23.6)	0.0220	0.66	0.37–1.19	0.51	0.21–1.24
Complaint of customers	23 (35.9)	50 (49.5)	64 (43.2)	137 (43.8)	0.2274	1.29	0.77–2.14	0.57	0.30–1.09
Short shelf-life and spoilage of foods	7 (10.9)	31 (30.7)	38 (25.7)	76 (24.3)	0.0135	1.28	0.73–2.25	0.28 *	0.11–0.68

^1^ *p*-value from chi-square test according to the stage of change. ^2^ OR; Odds Ratio, CI; Confidence Interval. * Independently significant in multivariate logistic regression models adjusted for age and operating system of restaurant (*p* < 0.05). ^3^ *N* (%): the number and response rate of ‘yes’ to each item. MA; Maintenance and Action stages, P; Preparation stage, PP; Pre-Preparation stage (contemplation and precontemplation stages).

**Table 4 nutrients-13-04375-t004:** Feasibility of actions regarding sodium reduction in restaurants according to the stages of behavioral change.

	MA (*n* = 64)	P (*n* = 101)	PP (*n* = 148)	Total (*n* = 313)	*p*-Value ^1^	P to Reference PP		MA to Reference P	
						OR ^2^	95% CI	OR	95% CI
Purchase									
Purchase foods after comparing sodium content in nutrition labels	7 (10.9) ^3^	8 (7.9)	22 (14.9)	37 (11.8)	0.2420	0.49	0.21–1.16	1.43	0.49–4.15
Score	3.3 ± 1.2 ^4^	3.4 ± 1.0	3.4 ± 1.2	3.4 ± 1.1	0.9410				
Cooking									
Measure the amount of salt and salty condiments while cooking	25 (39.1)	29 (28.7)	50 (33.8)	104 (33.2)	0.3809	0.79	0.46–1.37	1.59	0.82–3.09
Use salimeter and keep the standard salinity	24 (37.5)	23 (22.8)	31 (21.0)	78 (24.9)	0.0316	1.12	0.60–2.05	2.04 *	1.02–4.05
Cook with low-sodium sauce or broth	32 (50.0)	31 (30.7)	45 (30.4)	108 (34.5)	0.0139	1.01	0.59–1.76	2.26 *	1.18–4.31
Use herbs and spices to flavor dishes	32 (50.0)	33 (32.7)	48 (32.4)	113 (36.1)	0.0344	1.01	0.59–1.74	2.06 *	1.08–3.92
Apply cooking skills to make low- sodium dishes	32 (50.0)	21 (20.8)	31 (21.0)	84 (26.8)	<0.0001	0.99	0.53–1.85	3.81 ***	1.92–7.57
Subtotal average score	2.5 ± 0.9	2.9 ± 0.6	3.0 ± 0.9	2.9 ± 0.8	0.0005				
Menu adjustment									
Offer less salty options to the consumers	20 (31.3)	25 (24.8)	38 (25.7)	83 (26.5)	0.6216	0.95	0.53–1.71	1.38	0.69–2.77
Add low-salt menu	18 (28.1)	19 (18.8)	27 (18.2)	64 (20.5)	0.2314	1.04	0.54–1.99	1.69	0.81–3.54
Offer side menu with fresh vegetables and fruits	30 (46.9)	39 (38.5)	56 (37.8)	125 (39.9)	0.4427	1.03	0.61–1.74	1.40	0.74–2.64
Subtotal average score	2.7±1.1	3.0 ± 0.8	3.1 ± 0.9	3.0 ± 0.9	0.0950				
Serving									
Serve sauce separately	31 (48.4)	46 (45.5)	45 (30.4)	122 (39.0)	0.0122	1.91 *	1.13–3.24	1.12	0.60–2.10
Avoid serving salt-fermented foods as side dishes	40 (62.5)	53 (52.5)	77 (52.0)	170 (54.3)	0.3366	1.02	0.61–1.69	1.51	0.80–2.86
Serve less salty kimchi	39 (60.9)	48 (47.5)	61 (41.2)	148 (47.3)	0.0306	1.29	0.78–2.15	1.72	0.91–3.25
Serve small portions of kimchi	33 (51.6)	45 (44.6)	66 (44.6)	144 (46.0)	0.6066	1.00	0.60–1.66	1.33	0.71–2.48
Subtotal mean score	2.2 ± 0.9	2.4 ± 0.9	2.6 ± 1.0	2.5 ± 1.0	0.1096				
Offering information									
Display sodium content of foods served	14 (21.9)	19 (18.8)	32 (21.6)	65 (20.8)	0.8403	0.84	0.45–1.58	1.21	0.56–2.62
Put up promotional materials for reducing sodium intake	37 (57.8)	40 (39.6)	68 (46.0)	145 (46.3)	0.0728	0.71	0.46–1.29	2.09 *	1.11–3.95
Subtotal average score	2.7 ± 1.1	3.0 ± 1.0	2.9 ± 1.1	2.9 ± 1.1	0.1547				
Total average score	2.6 ± 0.8	2.9 ± 0.6	2.9 ± 0.7	2.8 ± 0.7	0.0262				

^1^ *p*-value from chi-square test or ANCOVA according to the stage of change. ^2^ OR; Odds Ratio, CI; Confidence Interval. * Independently significant in multivariate logistic regression models adjusted for age and operating system of restaurant (* *p* < 0.05, *** *p* < 0.001). ^3^ *N* (%): the number and response rate of ‘easy’ and ‘very easy’ to each item. ^4^ Mean ± SD: Mean score value of each section (score range = 1–5; 5; very difficult, 4; difficult, 3; neither difficult nor easy, 2; easy, 1; very easy); a lower score means more feasible to practice specific action regarding sodium reduction in restaurants. MA; Maintenance and Action stages, P; Preparation stage, PP; Pre-Preparation stage (contemplation and precontemplation stages).

**Table 5 nutrients-13-04375-t005:** Support needs to reduce sodium use in restaurants according to the stages of behavioral change.

	MA (*n* = 64)	P (*n* = 101)	PP (*n* = 148)	Total (*n* = 313)	*p*-Value ^1^	P to Reference PP		MA to Reference P	
						OR ^2^	95% CI	OR	95% CI
Willing to participate sodium reduction in restaurant under the support of government									
Yes	52 (83.9) ^3^	85 (88.5)	109 (76.2)	246 (81.7)	0.0479	2.41 *	1.15–5.04	0.67	0.27–1.69
Support needs									
Publicize participating restaurants	33 (51.6)	50 (49.5)	72 (48.7)	155 (49.5)	0.9269	1.04	0.62–1.72	1.09	0.58–2.03
Providing plaques of ‘healthy restaurant’ to participating restaurant	44 (68.8)	61 (60.4)	76 (51.4)	181 (57.8)	0.0511	1.45	0.87–2.41	1.44	0.74–2.80
Providing salimeter and educating how to use	29 (45.3)	59 (58.4)	71 (48.0)	159 (50.8)	0.1663	1.52	0.91–2.54	0.59	0.31–1.11
Analyzing sodium content and support to display it	16 (25.0)	42 (41.6)	59 (39.9)	117 (37.4)	0.0691	1.07	0.64–1.80	0.47 *	0.24–0.93
Training cooking skills to reduce salt use	29 (46.0)	50 (49.5)	84 (56.8)	163 (52.2)	0.2885	0.75	0.45–1.24	0.87	0.46–1.63
Support for educating employees	11 (17.2)	28 (27.7)	36 (24.3)	75 (24.0)	0.3002	1.19	0.67–2.12	0.54	0.25–1.18

^1^*p*-value from chi-square test according to the stage of change. ^2^ OR; Odds Ratio, CI; Confidence Interval. * Independently significant in multivariate logistic regression models adjusted for age and operating system of restaurant (*p* < 0.05). ^3^ *N* (%): the number and response rate of ‘yes’ to each item. MA; Maintenance and Action stages, P; Preparation stage, PP; Pre-Preparation stage (contemplation and precontemplation stages).

## Data Availability

Data are available from Kim, H.-K upon reasonable request.

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
