# Peer review of "Perceptions and Feasibility of Actions Related to Sodium Reduction among Restaurant Owners and Cooks in Seongnam, South Korea: Comparison According to Stages of Behavioral Change"

_nutrients, 2021, doi:10.3390/nu13124375_

Round 1

Reviewer 1 Report

This manuscript evaluates the perceptions and feasibility of action related to sodium reduction among restaurant owners and cooks in South Korea. The authors found a minority of restaurants in an active or maintenance phase of change with respect to lowering sodium and demonstrated differences in stage of change with respect to recognition of social environment, practice of assessing sodium content, and feasibility of actions. While the study had a reasonable sample size, it is entirely based upon self-reported survey results and descriptive in nature, with some additional concerns regarding the study design.

  1. The significance of this study is reduced with little information provided about baseline knowledge of the participants, particularly since the conclusion is that government education and resources improve compliance with lowering sodium, which has already been demonstrated in other studies to be effective. Even if specific knowledge of sodium was not assessed, the introduction should include a discussion of governmental efforts involved in the sodium reduction efforts to enable better context of restaurant changes. What government support and education was being provided to restaurants at the time of the study? Are changes for restaurants voluntary or compulsory? Is change willingness based upon knowledge level?
  2. The study was based on cross-sectional surveys of restaurant owners and cooks, collected at restaurant-associated events hosted by the local Food Safety Department over a 3 month period, with a sample size of 315. It is unclear what proportion of attendees this represents, so it is unclear as to whether this is a representative sample, and whether there was any possibility that the same restaurant could have been included more than once. Are these events obligatory for local restaurants? If not, there may be bias based upon attendees, with further bias based upon those who volunteered to participate in the study.
  3. The survey is entirely self-reported, with many subjective questions, and is modified from those used in prior studies without demonstrated validation that responses are reflective of true conditions. It is highly possible that restaurant owners and cooks have a response bias toward action. While the authors mention this, they should point out the direction of the bias, as this highlights the need for increased interventions.

Author Response

Authors would like to thank the reviewers for helpful comments and suggestions. We have revised manuscript addressing the issues reviewers have raised. Our point-by-point responses to the critiques are provided below. The revisions made are highlighted in the revised manuscript using the “track change”

Reviewer1

This manuscript evaluates the perceptions and feasibility of action related to sodium reduction among restaurant owners and cooks in South Korea. The authors found a minority of restaurants in an active or maintenance phase of change with respect to lowering sodium and demonstrated differences in stage of change with respect to recognition of social environment, practice of assessing sodium content, and feasibility of actions. While the study had a reasonable sample size, it is entirely based upon self-reported survey results and descriptive in nature, with some additional concerns regarding the study design.

  1. The significance of this study is reduced with little information provided about baseline knowledge of the participants, particularly since the conclusion is that government education and resources improve compliance with lowering sodium, which has already been demonstrated in other studies to be effective. Even if specific knowledge of sodium was not assessed, the introduction should include a discussion of governmental efforts involved in the sodium reduction efforts to enable better context of restaurant changes. What government support and education was being provided to restaurants at the time of the study? Are changes for restaurants voluntary or compulsory? Is change willingness based upon knowledge level?

→ Thank you for the suggestions. As the reviewer suggested, we have included governmental efforts for sodium reduction in restaurant food in the introduction (line 58-65). Government promoted ‘Healthier Restaurant’ program that publicize restaurants that offer considerable number of low-sodium meals or sodium-reduced menus. In addition, some local governments implemented customized programs to help reduce sodium reduction, such as low-sodium cooking consultation and training on the use of salimeter and the mobile application for salinity management. Restaurants could participate in the programs voluntarily. It was reported that the programs had a positive effect to reduce sodium content of restaurant menu items, but the factors to drive restaurant be willing to change have not been investigated.

2. The study was based on cross-sectional surveys of restaurant owners and cooks, collected at restaurant-associated events hosted by the local Food Safety Department over a 3-month period, with a sample size of 315. It is unclear what proportion of attendees this represents, so it is unclear as to whether this is a representative sample, and whether there was any possibility that the same restaurant could have been included more than once. Are these events obligatory for local restaurants? If not, there may be bias based upon attendees, with further bias based upon those who volunteered to participate in the study.

→ The process of survey was supplemented in the revised manuscript (line 92-95). Participants of the survey was recruited in mandatory annual hygiene training for restaurants and the participants attended the training program on behalf of the restaurant for which they worked. Thus, the same restaurant cannot be included in duplicate. Seongnam is a large city in Korea with residents about one million and many IT companies. The size and regional distribution of restaurants surveyed were similar to the restaurant’s status data in Seongnam city (https://www.data.go.kr/data/15076265/fileData.do), which has been added in the revised manuscript (line 173-174). In this regard, it seems that the restaurants surveyed are a representative sample, although not all attendees participate the survey due to busy schedule and informed consent.

3. The survey is entirely self-reported, with many subjective questions, and is modified from those used in prior studies without demonstrated validation that responses are reflective of true conditions. It is highly possible that restaurant owners and cooks have a response bias toward action. While the authors mention this, they should point out the direction of the bias, as this highlights the need for increased interventions.

Authors agree with the reviewer’s comment. As the reviewer suggested, we have indicated the direction of bias toward willingness and action for reducing sodium in restaurant owners and cooks, which suggests the need for increased intervention (line 414-417).   

Reviewer 2 Report

Comments for authors

In their paper, Ahn. et al aimed to assess the stages of behavioral change for sodium reduction in restaurant staff and owners in South Korea and to address perception differences between volunteers among each stage. 313 owners and cooks completed a questionnaire regarding behavioral change, practices and perceptive factors about sodium reduction in restaurants. The authors conclude that only 20% of total participants were in the action and maintenance stage for sodium reduction; however, since more than 80% of participants are willing to try with the appropriate assistance, motivational plans and promotion strategies are needed. This is an interesting paper; however certain issues need to be addressed.

Major issues:

  • Despite the fact that the authors state that they developed a new questionnaire with few modifications from those used in previous studies, they present questions of the questionnaire partially and the whole questionnaire is not presented. It would be very helpful to include it as a figure.
  • In the Methods section, the authors are thoroughly presenting the way that they separated participants in the 5 different stages. It would be easier for the readers to view a table or a figure representing the stages classification.
  • In the Methods section, please state any possible confounders used in the conducted analyses.
  • This paper needs improvement of language and extensive linguistic corrections.

Minor issues:

  • Abstract: Please be more specific about the study’s design.
  • Abstract, line 16: Please change “factors related to reducing sodium use in the restaurant” to “factors related to sodium reduction in restaurant meals”.
  • Introduction, line 31: Please add the word “intake” after “High dietary sodium”.
  • Introduction, line 46: Please rephrase the following sentence “With the increased consumption of foods cooked outside the home, the contribution of restaurant foods to dietary sodium intake has become significant. It was reported that restaurant foods ...” to “With the increased consumption of meals cooked outside the home, contribution of restaurant food to dietary sodium intake has become significant. It was reported that restaurant meals ...”.
  • Methods, line 120: Please change “barriers to practicing” to “barriers in practicing”.
  • Methods, line 121: Please change “barriers to reducing” to “barriers in reducing”.
  • Methods, line 141: Please change “The results are shown” to “Results are presented”
  • Results, line 149: Please provide standard deviation for mean age of participants.
  • Results, line 215: the authors probably mean “PP stage”.
  • Table 2: Please explain what superscript “b” stands for.

Table 4: The symbol “**” cannot be found in the table. Please add it in the table or remove it from the abbreviations 

Author Response

Authors would like to thank the reviewers for helpful comments and suggestions. We have revised manuscript addressing the issues reviewers have raised. Our point-by-point responses to the critiques are provided below. The revisions made are highlighted in the revised manuscript using the “track change”

Reviewer2

In their paper, Ahn. et al aimed to assess the stages of behavioral change for sodium reduction in restaurant staff and owners in South Korea and to address perception differences between volunteers among each stage. 313 owners and cooks completed a questionnaire regarding behavioral change, practices and perceptive factors about sodium reduction in restaurants. The authors conclude that only 20% of total participants were in the action and maintenance stage for sodium reduction; however, since more than 80% of participants are willing to try with the appropriate assistance, motivational plans and promotion strategies are needed. This is an interesting paper; however certain issues need to be addressed.

Major issues:

  • Despite the fact that the authors state that they developed a new questionnaire with few modifications from those used in previous studies, they present questions of the questionnaire partially and the whole questionnaire is not presented. It would be very helpful to include it as a figure.

Authors agree with the reviewer’s comment, but we think that it would be better to place it in supplements rather than a regular figure. The questionnaire used has been attached as ‘supplementary files’.

  • In the Methods section, the authors are thoroughly presenting the way that they separated participants in the 5 different stages. It would be easier for the readers to view a table or a figure representing the stages classification.

→ As the reviewer suggested, stage classification process has been presented as a figure in the revised manuscript (Fig 1).

  • In the Methods section, please state any possible confounders used in the conducted analyses.

→ The possible confounders in this study have been stated in statistical analysis section (line 154-156) in the revised manuscript.

  • This paper needs improvement of language and extensive linguistic corrections.

→ The manuscript has been reviewed by a native speaker with academic background in English language editing service agency (Essayreview, contact information: [email protected]) for grammar and syntax and revised.

Minor issues:

  • Abstract: Please be more specific about the study’s design.

→ The study’s design has been rephrased in the revised manuscript (line 14-17).

  • Abstract, line 16: Please change “factors related to reducing sodium use in the restaurant” to “factors related to sodium reduction in restaurant meals”.

→ We have changed the sentence as suggested (line 16-17).

  • Introduction, line 31: Please add the word “intake” after “High dietary sodium”.

→ The word ‘intake’ has been added in the revised manuscript (line 33).

  • Introduction, line 46: Please rephrase the following sentence “With the increased consumption of foods cooked outside the home, the contribution of restaurant foods to dietary sodium intake has become significant. It was reported that restaurant foods ...” to “With the increased consumption of meals cooked outside the home, contribution of restaurant food to dietary sodium intake has become significant. It was reported that restaurant meals ...”.

→ The sentence has been changed as the reviewer suggested (line 47-50).

  • Methods, line 120: Please change “barriers to practicing” to “barriers in practicing”.

→ We have changed the phrase as the reviewer suggested (line 137).

  • Methods, line 121: Please change “barriers to reducing” to “barriers in reducing”.

→ We have changed the phrase in the whole manuscript (line 228, 229, 238, 279) as well as the line you mentioned (line 139).

  • Methods, line 141: Please change “The results are shown” to “Results are presented”

→ The sentence has been changed as the reviewer suggested (line 160).

  • Results, line 149: Please provide standard deviation for mean age of participants.

→ The standard deviation for mean age of participants has been added in line168.

  • Results, line 215: the authors probably mean “PP stage”.

→ Thank you for finding our mistake. We corrected the mistake in the revised manuscript (line 236).

  • Table 2: Please explain what superscript “b” stands for.

→ Table 2 does not have superscript. It seems that the comment is related with Table 1. The superscripts have been explained in the footnotes of Table 1 (214-216).

  • Table 4: The symbol “**” cannot be found in the table. Please add it in the table or remove it from the abbreviations 

→ The symbol “**” has been deleted from the footnote in Table 4 (line 287).
